# Genetic depletion studies inform receptor usage by virulent hantaviruses in human endothelial cells

Maria Eugenia Dieterle[1], Carles Solà-Riera[2], Chunyan Ye[3], Samuel M Goodfellow[3], Eva Mittler[1], Ezgi Kasikci[1], Steven B Bradfute[3], Jonas Klingström[2], Rohit K Jangra[1]*, Kartik Chandran[1]*

[1]Department of Microbiology and Immunology, Albert Einstein College of Medicine, Bronx, United States; [2]Center for Infectious Medicine, Department of Medicine Huddinge, Karolinska Institutet, Stockholm, Sweden; [3]University of New Mexico Health Science Center, Center for Global Health, Department of Internal Medicine, Albuquerque, United States

**Abstract** Hantaviruses are RNA viruses with known epidemic threat and potential for emergence. Several rodent-borne hantaviruses cause zoonoses accompanied by severe illness and death. However, assessments of zoonotic risk and the development of countermeasures are challenged by our limited knowledge of the molecular mechanisms of hantavirus infection, including the identities of cell entry receptors and their roles in influencing viral host range and virulence. Despite the long-standing presumption that β3/β1-containing integrins are the major hantavirus entry receptors, rigorous genetic loss-of-function evidence supporting their requirement, and that of decay-accelerating factor (DAF), is lacking. Here, we used CRISPR/Cas9 engineering to knockout candidate hantavirus receptors, singly and in combination, in a human endothelial cell line that recapitulates the properties of primary microvascular endothelial cells, the major targets of viral infection in humans. The loss of β3 integrin, β1 integrin, and/or DAF had little or no effect on entry by a large panel of hantaviruses. By contrast, loss of protocadherin-1, a recently identified entry receptor for some hantaviruses, substantially reduced hantavirus entry and infection. We conclude that major host molecules necessary for endothelial cell entry by PCDH1-independent hantaviruses remain to be discovered.

*For correspondence:
rohit.jangra@einsteinmed.org (RKJ);
kartik.chandran@einsteinmed.org (KC)

## Introduction

Hantaviruses are a large family of enveloped viruses with segmented negative-strand RNA genomes whose members infect a wide range of mammalian hosts (*Elliott et al., 1991*; *Mittler et al., 2019*; *Vaheri et al., 2013*). The zoonotic transmission of some rodent-borne hantaviruses is associated with two major diseases in humans, hemorrhagic fever with renal syndrome (HFRS), and hantavirus cardiopulmonary syndrome (HCPS), in endemic regions of Europe/Asia and North/South America, respectively (*Peters et al., 1999*). Zoonotic infections by some other hantaviruses (e.g., Seoul virus, SEOV) have a global distribution mirroring that of their rodent hosts. Most HFRS cases are associated with the 'Old-World hantaviruses' Hantaan virus (HTNV), SEOV, Dobrava-Belgrade virus (DOBV), and Puumala virus (PUUV) infections, whereas HCPS is primarily caused by the 'New-World hantaviruses' Andes virus (ANDV) and Sin Nombre virus (SNV) (*Jiang et al., 2016*; *Macneil et al., 2011*). Although humans are typically dead-end hosts for rodent-borne hantaviruses, several incidents of ANDV person-to-person transmission, including a recent superspreader event, have been documented, underscoring the epidemic threat posed by these agents (*Martínez et al., 2020*; *Padula et al., 1998*). Despite this strong potential for transmission from known and unknown host

reservoirs and the significant burden of severe disease, no FDA-approved vaccines or specific therapeutics are available for hantaviruses (*Mittler et al., 2019*). Both assessments of viral zoonotic risk and the development of antiviral treatments are challenged by our limited understanding of the molecular mechanisms of hantavirus infection, including the identities, viral and host species specificities, and functional roles of cellular entry receptors.

Integrins (αVβ3, α5β1, αMβ2, and αXβ2) and components of the complement system (e.g., decay-accelerating factor [DAF]) have been previously proposed as candidate hantavirus receptors and/or entry factors, based largely on cDNA complementation experiments to rescue infection in nonpermissive cell lines, receptor-blocking studies in non-polarized and polarized endothelial cells and/or binding assays (*Buranda et al., 2010*; *Gavrilovskaya et al., 1999*; *Gavrilovskaya et al., 1998*; *Krautkrämer and Zeier, 2008*; *Popugaeva et al., 2012*; *Raftery et al., 2014*). β3- and β1-containing integrins have been presumed to be the major entry receptors for virulent and avirulent hantaviruses, respectively, for over two decades (*Gavrilovskaya et al., 1999*; *Gavrilovskaya et al., 2002*; *Gavrilovskaya et al., 1998*; *Raymond et al., 2005*). More recently, we identified protocadherin-1 (PCDH1) as a critical determinant of attachment, entry, and infection by New-World hantaviruses, but not their Old-World counterparts, in primary human microvascular endothelial cells (*Jangra et al., 2018*), which are major targets of hantavirus infection in vivo (*Mackow and Gavrilovskaya, 2009*). However, evidentiary support for each of the above putative receptors differs substantially. *PCDH1* was identified in a comprehensive genetic screen for ANDV entry factors, and its genetic depletion in endothelial cells inhibited viral entry and infection. Further, genetic loss of *PCDH1* reduced viral multiplication in Syrian hamsters and protected them from lethal HCPS-like disease (*Jangra et al., 2018*). By contrast, none of the other candidate receptors were observed as hits in two independently conducted, unbiased genetic screens for hantavirus host factors (*Jangra et al., 2018*; *Kleinfelter et al., 2015*; *Petersen et al., 2014*). Moreover, to our knowledge, they have not been rigorously evaluated for their genetic requirement in physiologically relevant in vitro and in vivo models. To date, therefore, the critical host determinants of hantavirus entry remain incompletely defined.

Here, we used a genetic depletion/complementation strategy to investigate the requirements for four hantavirus receptor candidates/entry host factors described in the literature – β3 integrin, β1 integrin, DAF, and PCDH1 – during glycoprotein-mediated entry and infection of human endothelial cells by divergent hantaviruses.

## Results and discussion

We first selected a previously well-characterized human microvascular endothelial cell line, TIME, as a biologically faithful and genetically manipulable model for primary endothelial cells (*Venetsanakos et al., 2002*). We confirmed through cell staining and flow cytometry that this cell line resembled primary human umbilical vein endothelial cells (HUVEC) in expressing key endothelial markers – the platelet endothelial cell adhesion molecule-1 (PECAM-1/CD31) and the von Willebrand factor (vWF) – as well as the four candidate hantavirus receptors under investigation (*Figure 1a*). Furthermore, TIME cells were susceptible to hantavirus entry mediated by divergent Gn/Gc proteins (*Figure 2*). We concluded, therefore, that TIME cells were also suitable to study the virus–host interactions that drive hantavirus entry in endothelial cells.

We used CRISPR/Cas9 genome engineering to knockout (KO) *PCDH1*, *DAF*, and the genes encoding β3 and β1 integrins (*ITGB3* and *ITGB1*, respectively), in TIME cells. Subpopulations of CRISPR/Cas9-engineered, receptor-deficient cells were isolated by FACS following staining of live cells with protein-specific antibodies. Gene KO was verified by Sanger sequencing of PCR amplicons bearing the targeted genomic loci (*Figure 1—figure supplement 1* ). Loss of protein expression at the cell surface and in cell extracts was verified by flow cytometry and western blotting, respectively (*Figure 1b,c*).

We next exposed WT and KO TIME cells to recombinant vesicular stomatitis viruses carrying the Gn/Gc glycoproteins of ANDV, SNV, and HTNV (rVSV-Gn/Gc) (*Figure 2a,b*). A similar surrogate virus bearing the Ebola virus glycoprotein (rVSV-EBOV GP) was used as a negative control. *PCDH1* KO substantially diminished ANDV and SNV Gn/Gc-mediated infection, whereas it had no discernible effect on that by HTNV Gn/Gc. Unexpectedly, *ITGB3* and *DAF* KOs did not inhibit endothelial cell entry mediated by these glycoproteins (*Figure 2a,b*). To examine the effect of candidate receptor

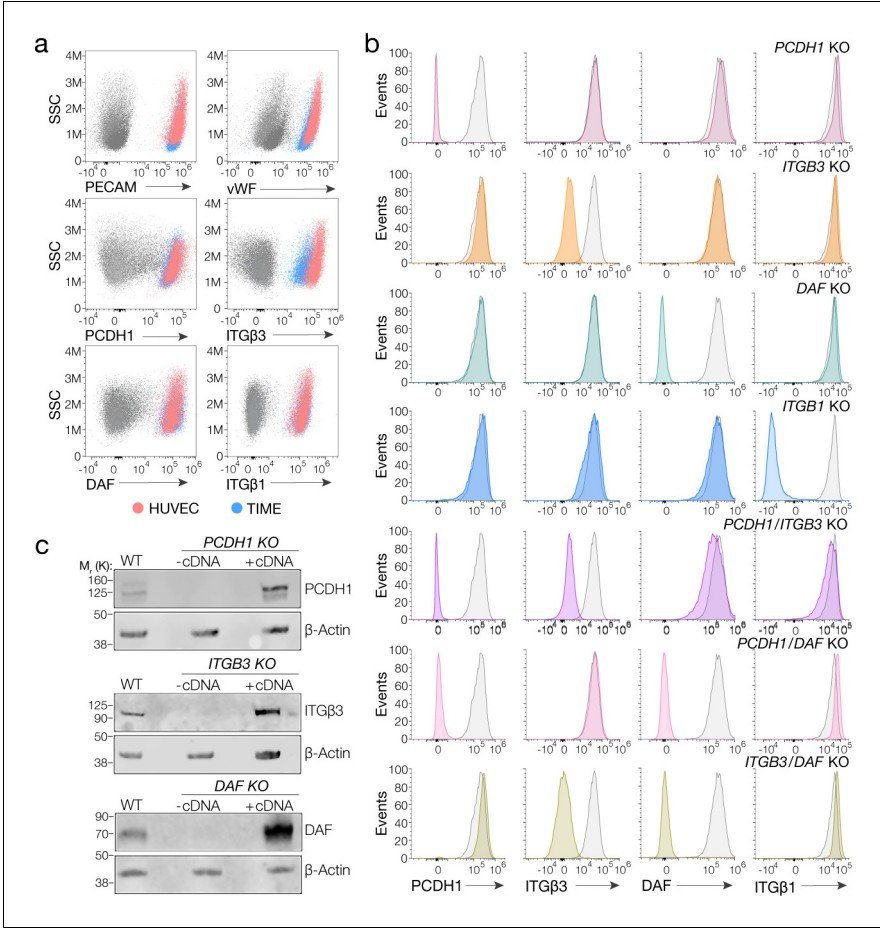

**Figure 1.** Suitability of TIME cells as a model to study hantavirus entry and the generation of knockout cells. (**a**) Upper panels, total flow cytometry plots of HUVEC and TIME cells stained for endothelial cell markers PECAM and von Willebrand factor (vWF). Medium and lower panels, surface flow cytometry plots of HUVEC and TIME cells stained for PCDH1, β3 integrin, DAF, β1 integrin. (**b**) Surface flow cytometry of wild-type (WT) and knockout (KO) TIME cells stained as above. Histograms of WT cells are shown in gray; single- and double-KO cells are shown in color. (**c**) Western blot analysis of WT TIME cells and KO cells ± cDNA. β-Actin was used as a loading control. The online version of this article includes the following source data and figure supplement(s) for figure 1:

**Source data 1.** Original blot of WT TIME cells and KO cells ± cDNA.
**Figure supplement 1.** Sanger sequences retrieved from the targeted genomic loci for each knockout cell population.

loss on a larger panel of genetically divergent hantaviruses with varied levels of known virulence in humans, we extended these studies to rVSVs bearing Gn/Gc proteins from the New-World hantaviruses Choclo virus (CHOV), Maporal virus (MPRLV), Prospect Hill virus (PHV), and the Old-World hantaviruses PUUV, SEOV, and DOBV (*Figure 2c*). Loss of *PCDH1* substantially reduced endothelial cell entry and infection mediated by the glycoproteins from the New-World viruses but not the Old-World viruses, in a manner that could be restored by complementation with *PCDH1* cDNA. By contrast, and as observed above for ANDV, SNV, and HTNV, none of these viruses displayed a requirement for *ITGB3* or *DAF* (*Figures 1c–2c*). These findings confirm and extend the critical role played by PCDH1 in cell entry by New-World hantaviruses. They are also inconsistent with the hypotheses that β3 integrin and DAF are necessary for hantavirus cell entry in endothelial cells.

To account for potential receptor redundancy and cross-talk, we next generated and evaluated double-KO populations of TIME cells through CRISPR/Cas9 engineering (*Figure 1b*). *PCDH1/ITGB3* and *PCDH1/DAF1* KO cells resembled *PCDH1* KO cells in susceptibility to rVSV-Gn/Gc infection (*Figure 2d*), indicating that the loss of PCDH1 did not unmask viral requirements for β3 integrin or

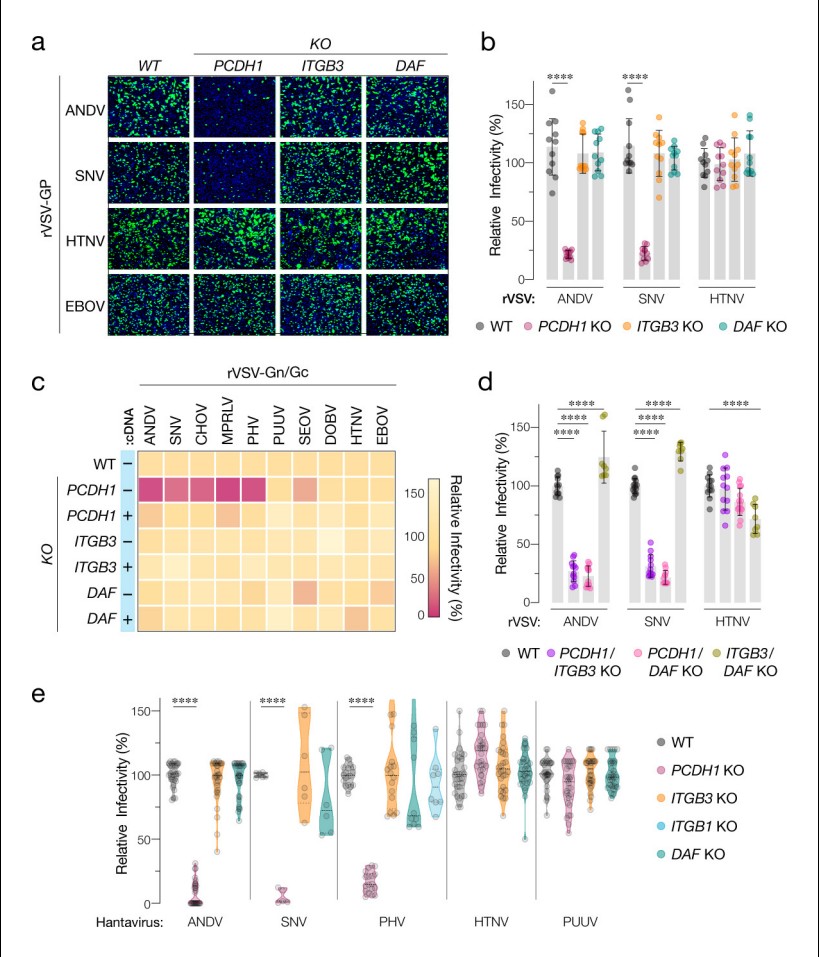

**Figure 2.** Hantavirus receptor requirement in endothelial cells. (**a**) Representative images of eGFP-positive rVSV-infected wild-type (WT) and *PCDH1*, *ITGB3*, and DAF knockout (KO) TIME cells. Nuclei were stained with Hoechst (blue). (**b**) WT and KO cells were exposed to the indicated rVSV-Gn/Gc. n=11 for each cell line from three independent experiments. WT versus KO cells, two-way ANOVA with Dunnett's test; ***p<0.0001. Other comparisons were not statistically significant (p>0.05). (**c**) WT and KO cells lacking (–cDNA) or expressing the corresponding cDNA (+cDNA) were exposed to rVSVs bearing the indicated hantavirus glycoproteins. Viral infectivities are shown in the heatmap. Averages are from three independent experiments (**d**) WT and double-KO cells were exposed to rVSV-Gn/Gc. *PCDH1/ITGB3* KO, n = 12; *PCDH1/DAF* KO, n = 12; and *ITGB3/DAF* KO, n=9 from three independent experiments. WT versus KO cells, two-way ANOVA with Dunnett's test; ****p<0.0001. Other comparisons were not statistically significant. (**e**) Cells were exposed to authentic hantaviruses and infected cells were manually enumerated by immunofluorescence microscopy for ANDV, HTNV, and PUUV (each point represents infectivity of the average of positive cells per field relative to WT). Data are from two independent experiments. PHV- and SNV-infected cells were detected and enumerated by automated imaging following immunofluorescence staining. For PHV: WT and *PCDH1* KO n=18, *ITGB3* KO n = 16 from four independent experiments; *DAF* KO, n=10, *ITGB1* KO n=8, from three independent experiments. For SNV: WT, *ITGB3* KO, *DAF* KO n=6; *PCDH1* KO n = 5 from three independent experiments. Averages ± SD are shown. WT versus KO cells, two-way ANOVA with Tukey's test; ***p<0.0001. Other comparisons were not statistically significant (p>0.05). The online version of this article includes the following figure supplement(s) for figure 2:

**Figure supplement 1.** Dispensability of β1 integrin in TIME cells.

---

DAF, or vice versa. Furthermore, *ITGB3/DAF* KO cells remained fully susceptible to viral entry (ANDV, SNV) or exhibited only a modest (~25%) reduction (HTNV) (*Figure 2d*), suggesting that the lack of viral entry phenotypes in the *ITGB3* and *DAF* single-KO cells cannot be explained by the functional redundancy of these two proteins.

Finally, we evaluated the single-KO TIME cells in infections with authentic hantaviruses (*Figure 2e*). These experiments independently corroborated our results with rVSVs. Specifically, we observed a critical role for PCDH1 in endothelial cell infection by the New-World hantaviruses ANDV, SNV, and PHV, and no apparent roles for β3 integrin and DAF in entry by virulent New-World and Old-World hantaviruses. Concordantly, quantitative real-time PCR revealed a substantial reduction in the generation of SNV progeny genomes at 24 hr post-infection in cells deficient for PCDH1 (5 ± 2%) but not β3 integrin (108 ± 15%) or DAF (82 ± 13%; mean ± SEM, n=6 from two independent experiments for each cell line). Contrary to earlier hypotheses suggesting β1 integrin as receptors for avirulent hantaviruses (*Gavrilovskaya et al., 1998*), we also found no apparent role for β1 integrin in entry by both the rVSV and the authentic version of New-World hantavirus PHV (*Figure 2e*, *Figure 2—figure supplement 1*).

Endothelial cells are primary targets of the viral infection and subversion that is central to hantavirus disease in humans. Of the major receptor/entry host factor candidates proposed to date for hantaviruses (β3 integrin, β1 integrin, DAF, PCDH1), only the genetic loss of *PCDH1* was associated with the reduction of viral infection in endothelial cells in this study, underscoring PCDH1's critical role as an entry receptor for New-World hantaviruses, but not their Old-World counterparts in these cells. We conclude that the other previously described candidate receptors (β3 integrin, β1 integrin, DAF) are dispensable for viral entry in endothelial cells. We note that our results do not rule out that one or more of these proteins is involved in (1) hantavirus entry into other cell types not examined herein; (2) hantavirus entry into polarized cells, as proposed for DAF (*Krautkrämer and Zeier, 2008*); or (3) endothelial cell subversion post-viral entry, as shown previously (*Gavrilovskaya et al., 1999*; *Gavrilovskaya et al., 1998*; *Krautkrämer and Zeier, 2008*). The sum of the evidence does indicate, however, that the major host molecules necessary for entry and infection in endothelial cells by the PCDH1-independent, HFRS-causing Old-World hantaviruses remain to be discovered.

## Materials and methods

### Cells

HUVEC (C2517A-Lonza) and TIME cells (ATCC CRL4025) were cultured as described previously (*Venetsanakos et al., 2002*). HUVEC and TIME cells were cultured in EBM-2 Basal Medium (CC-3156 Lonza) and EGM-2 SingleQuotsTM Supplements (CC-4176-Lonza) supplemented with 5% heat-inactivated fetal bovine serum (FBS, Atlanta Biologicals), 1% penicillin/streptomycin (P/S, GIBCO), and 1% Gluta-MAX (GIBCO). Cells were passaged every 3–4 days using 0.05% Trypsin/EDTA solution (GIBCO). African vervet monkey kidney Vero cells were maintained in Dulbecco's modified Eagle medium (DMEM) (high glucose) supplemented with 2% heat-inactivated FBS, 1% P/S, and 1% Gluta-MAX. 293T cells were cultured in DMEM (high glucose, GIBCO) supplemented with 10% heat-inactivated FBS (Atlanta Biologicals), 1% penicillin/streptomycin (P/S, GIBCO), and 1% Gluta-MAX (GIBCO). All cell lines used here were negative for mycoplasma. STR profiling was performed for TIME cell line ATCC CRL-4025 authentication (Genomics Core, Albert Einstein College of Medicine).

### Surface and total flow cytometry

For surface staining, cells were kept at 4°C. Foxp3/Transcription Factor Staining Buffer Kit was used for intracellular staining following the manufacturer's instructions (TNB-0607, KIT). Antibodies used were as follows: AF480-Rabbit-α–vWF (ab195028, Abcam), PeCy7-Mouse-α–PECAM (563651, BD), APC-Mouse-α–β1-Integrin (559883, BD), AF 647-Mouse-α–β3-Integrin (336407, Biolegends), PE-Mouse-α–DAF (555694, BD), and α–PCDH1 mAb 3305 (2)/AF488-α–Human (A-11013, Invitrogen). Zombie NIR Fixable Viability Kit (BioLegend) was used to assess live/dead status of the cells.

### Western blotting

Performed as described (*Jangra et al., 2018*). Briefly, cells were lysed in radioimmunoprecipitation assay lysis buffer (RIPA) containing protease inhibitors (Sigma). Cell extracts were normalized by Bradford assay, and 40 μg of protein was added per lane. The extracted proteins were separated by 10% SDS–PAGE and transferred onto 0.22 μm nitrocellulose membranes (GE Healthcare Life Sciences). Blots were cut approximately in half for subsequent incubation with primary antibodies and aligned back together for imaging. Antibodies used were as follows: Mouse α–PCDH1 (sc-

81816, Santa Cruz) 1:200; Rabbit–α–β3 Integrin (#4702, Cell Signaling) 1:300; Mouse-α–DAF (NaM16-4D3, Santa Cruz) 1:200; Mouse α–β-Actin (sc-4778, Santa Cruz) 1:300. IRDye 680LT Goat α-Rabbit IgG or IRDye 680LT Goat α-Mouse secondaries Abs (LI-COR) were used at a dilution of 1:10,000, and the final blot was then imaged using a LI-COR Fc fluorescent imager.

## Generation of KO cell populations

A CRISPR sgRNA was designed to target, PCDH1 5′-GTTTGAGCGGCCCTCCTATGAGG-3′ (*Jangra et al., 2018*); *ITGB3* 5′-CCACGCGAGGTGTGAGCTCCTGC-3′; *DAF* 5′-CCCCCAGATG TACCTAATGCCCA-3′; and *ITGB1* 5′-AATGTAACCAACCGTAGCAAAGG-3′ (protospacer acceptor motif [PAM] is underlined). sgRNAs were cloned into lentiCRISPR v2 (Addgene plasmid # 52961), and CRISPR-Cas9 engineering was performed as described (*Jangra et al., 2018*). Receptor-negative subpopulations of TIME cells were isolated by FACS sorting. Double-KO cell populations were generated using a single-KO cell population and then CRISPR-Cas9 engineering with a different sgRNA as described before. Double-receptor-negative subpopulations of TIME cells were isolated by FACS sorting. The targeted genomic loci in genomic DNA isolated from these subpopulations were amplified by PCR, and the amplicons were TA-cloned into the pGEM-T vector. Fifteen to 20 clones for each KO cell population were subjected to Sanger sequencing. For each KO cell population, all sequenced clones showed an indel at the targeted site, resulting in a frameshift that brought one or more stop codons into frame.

## Stable cell populations expressing PCDH1/ITGB3/DAF in KO cells

TIME KO cell populations ectopically expressing gene variants were generated by retroviral transduction. PCDH1 canonical isoform Q08174-1 (Uniprot), DAF canonical isoform P08174-1 (Uniprot), and ITGB3 canonical isoform P05106-1 (Uniprot) were codon optimized (humanized) and then cloned into pBABE-puro vector (Addgene plasmid # 1764). Retroviruses packaging the transgenes were produced by transfection in 293T cells (*Kleinfelter et al., 2015*), and target cells were exposed to sterile-filtered retrovirus supernatants in the presence of polybrene (8 µg/ml) at 36 hr and 60 hr post-transfection. Media was replaced 6 hr after each transduction with EBM-2 Basal Medium. Transgene expression was confirmed by FACS and western blot. Receptor-positive subpopulations of TIME cells were isolated by FACS sorting.

## rVSVs and infections

rVSVs expressing eGFP and bearing Gn/Gc from ANDV, SNV, HTNV, SEOV, DOBV, MPRLV, EBOV, PHV, or mNeon green phosphoprotein P fusion protein (mNG-P) bearing Gn/Gc PUUV were described previously (*Jangra et al., 2018*; *Kerkman et al., 2019*; *Kleinfelter et al., 2015*; *Slough et al., 2019*; *Wong et al., 2010*). rVSV expressing mNeon Green fused to the VSV phosphoprotein and bearing Gn/Gc from CHOV (GenBank accession number KT983772.1) was generated by using a plasmid-based rescue system in 293T human embryonic kidney fibroblast cells as described before (*Kleinfelter et al., 2015*). Rescued virus was amplified on Vero cells and its identity was verified by sequencing of the Gn/Gc-encoding gene. Viral infectivity was measured at 14 hr post-infection by automated enumeration of eGFP- or mNeongreen-positive cells using a Cytation5 automated fluorescence microscope (BioTek) and analyzed using the Gen5 data analysis software (BioTek). 100% relative infectivity corresponds to 18–35% infected cells for ANDV, SNV, HTNV, EBOV, CHOV, MPRLV and 8–15% for SEOV, DOBV, PHV, PUUV.

## Hantavirus and infections

ANDV isolate Chile-9717869, HTNV isolate 76–118, PUUV isolate Sotkamo, PHV, and SNV isolate SN77734 were used. TIME cells were seeded in 24-well plates (on glass coverslips) or 96-well plates and infected with a multiplicity of infection of 1. After 48 hr (24 hr for SNV), immunofluorescence was done as described before (*Stoltz et al., 2007*). Briefly, human polyclonal antibodies from convalescent patient serum plus anti-hantavirus nucleocapsid protein B5D9 monoclonal antibody (Progen) were used for HTNV, PUUV, ANDV. Rabbit polyclonal sera specific for HTNV nucleoprotein NR-12152 (BEI Resources) was used for PHV and SNV and incubated for an hour. For HTNV-, PUUV-, ANDV-infected cells, AF594-α–Human and α-Mouse IgG antibodies (Invitrogen) were used as secondaries. For SNV and PHV, AF488-α–Rabbit IgG was used (Invitrogen). Counterstaining of nuclei

was done with DAPI (Thermo Fisher Scientific). Cells were imaged by fluorescence microscopy (Nikon, Eclipse TE300) with a $60\times$ objective or by automated enumeration of eGFP-positive cells using a Cytation5 automated fluorescence microscope (BioTek) and analyzed using the Gen5 data analysis software (BioTek).

## SNV RNA levels

WT and KO TIME cells were harvested before infection (negative control) or at 24 hr after infection. Degenerate primers and probe were adopted from *Kramski et al., 2007* based on the S-segment of the SNV genome. Human β-actin primers with a VIC/TAMRA-dye probe (Applied Biosystems, Cat. # 4310881E) was used as an endogenous control. The qPCR were carried out by using TaqMan Fast Advanced Master Mix (Applied Biosystems, Thermo Fisher Scientific) according to the manufacturer's instructions. Relative gene fold change was calculated by normalizing SNV to β-actin in cells using ΔΔCt values. Averages ± SEM from two independent experiments are shown.

## Statistics and reproducibility

The number of independent experiments and the measures of central tendency and dispersion used in each case are indicated in the figure legends. The testing level (alpha) was 0.05 for all statistical tests. Statistical comparisons were carried out by two-way ANOVA with a post hoc correction for family-wise error rate. All analyses were carried out in GraphPad Prism.

# Acknowledgements

We thank Isabel Gutierrez, Estefania Valencia, Laura Polanco, and Sandra Diaz for laboratory management and technical support. This work was supported by NIH grant R01AI132633 (to KC), and Swedish Research Council grant 2018–02646 (to JK). MED was partially supported as a Latin American Fellow in the Biomedical Sciences of the Pew Charitable Trusts. SMG was partially supported by the UNM HSC Infectious Disease and Inflammation Program NIH training grant T32AI007538. The Einstein Flow Cytometry Core is partially supported by NIH grant P30CA013330.

# Additional information

### Competing interests

Rohit K Jangra: is named co-inventor on US patent 10,105,433 covering PCDH1 as a target for anti-hantavirus treatments. Kartik Chandran: is named co-inventor on US patent 10,105,433 covering PCDH1 as a target for anti-hantavirus treatments. K.C. is a member of the scientific advisory boards of Integrum Scientific, LLC; Biovaxys Technology Corp; and the Pandemic Security Initiative of Celdara Medical, LLC. The other authors declare that no competing interests exist.

### Funding

| Funder | Grant reference number | Author |
|---|---|---|
| National Institutes of Health | R01AI132633 | Kartik Chandran |
| Swedish Research Council | 2018-02646 | Jonas Klingstrom |
| National Institutes of Health | T32AI007538 | Samuel M Goodfellow |
| Pew Charitable Trusts | Latin American Fellow in the Biomedical Sciences | Maria Eugenia Dieterle |

The funders had no role in study design, data collection and interpretation, or the decision to submit the work for publication.

### Author contributions

Maria Eugenia Dieterle, Conceptualization, Data curation, Formal analysis, Investigation, Visualization, Methodology, Writing - original draft, Writing - review and editing; Carles Solà-Riera, Chunyan Ye, Samuel M Goodfellow, Eva Mittler, Ezgi Kasikci, Investigation, Writing - review and editing;

Steven B Bradfute, Supervision, Writing - review and editing; Jonas Klingström, Supervision, Funding acquisition, Writing - review and editing; Rohit K Jangra, Conceptualization, Investigation, Writing - original draft, Writing - review and editing; Kartik Chandran, Conceptualization, Supervision, Funding acquisition, Writing - original draft, Project administration, Writing - review and editing

### Author ORCIDs
Maria Eugenia Dieterle https://orcid.org/0000-0002-5257-7253
Ezgi Kasikci http://orcid.org/0000-0003-4570-3781
Jonas Klingström https://orcid.org/0000-0001-9076-1441
Rohit K Jangra https://orcid.org/0000-0002-3119-0869
Kartik Chandran https://orcid.org/0000-0003-0232-7077

### Decision letter and Author response
Decision letter https://doi.org/10.7554/eLife.69708.sa1
Author response https://doi.org/10.7554/eLife.69708.sa2

## Additional files

### Supplementary files
• Transparent reporting form

### Data availability
All data generated or analysed during this study are included in the manuscript and supporting files. Source data files have been provided for Figure 1.

The following previously published datasets were used:

| Author(s) | Year | Dataset title | Dataset URL | Database and Identifier |
|---|---|---|---|---|
| The UniProt Consortium | 2021 | UniProt: the universal protein knowledgebase in 2021 | https://www.uniprot.org/ | UniParc, www.uniprot. org/ |

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

# Appendix 1

**Appendix 1—key resources table**

| Reagent type (species) or resource | Designation | Source or reference | Identifiers | Additional information |
|---|---|---|---|---|
| Gene (*Homo sapiens*) | *PCDH1* | GenBank | Gene ID: 5097 | |
| Gene (*Homo sapiens*) | *ITGB3* | GenBank | Gene ID: 3690 | |
| Gene (*Homo sapiens*) | *DAF* | GenBank | Gene ID: 1604 | |
| Gene (*Homo sapiens*) | *ITGB1* | GenBank | Gene ID: 3688 | |
| Strain, strain background (virus) | rVSV eGFP ANDV Gn/Gc | *Kleinfelter et al., 2015* | | |
| Strain, strain background (virus) | rVSV eGFP SNV Gn/Gc | *Kleinfelter et al., 2015* | | |
| Strain, strain background (virus) | rVSV eGFP HTNV Gn/Gc | *Kleinfelter et al., 2015* | | |
| Strain, strain background (virus) | rVSV eGFP SEOV Gn/Gc | *Jangra et al., 2018* | | |
| Strain, strain background (virus) | rVSV eGFP DOBV Gn/Gc | *Slough et al., 2019* | | |
| Strain, strain background (virus) | rVSV eGFP MPRLV Gn/Gc | *Jangra et al., 2018* | | |
| Strain, strain background (virus) | rVSV eGFP PHV Gn/Gc | *Jangra et al., 2018* | | |
| Strain, strain background (virus) | rVSV mNeongreen-P PUUV-Gn/Gc | *Kerkman et al., 2019*, This study | | Laboratory of K. Chandran |
| Strain, strain background (virus) | rVSV mNeongreen-P CHOV Gn/Gc | This study | GenBank # KT983772.1 | Laboratory of K. Chandran/VSV antigenome plasmid (*Whelan et al., 1995*), plasmids expressing T7 polymerase and VSV N, P, M, G, and L (*Witko et al., 2006*) |
| Strain, strain background (virus) | rVSV-EBOV/Mayinga GP (EBOV/H.sap-tc/COD/76/Yambuku-Mayinga) | *Wong et al., 2010* | | |
| Strain, strain background (Hantavirus) | ANDV isolate Chile-9717869 | N/A | | |
| Strain, strain background (Hantavirus) | HTNV isolate 76–118 | N/A | | |
| Strain, strain background (Hantavirus) | PUUV isolate Sotkamo | N/A | | |

*Continued on next page*

*Appendix 1—key resources table continued*

| Reagent type (species) or resource | Designation | Source or reference | Identifiers | Additional information |
|---|---|---|---|---|
| Strain, strain background (Hantavirus) | SNV isolate SN77734 | *Botten et al., 2000* | | |
| Strain, strain background (Hantavirus) | PHV | N/A | | Laboratory of K. Chandran |
| Cell line (*H. sapiens*) | 293T | ATCC | Cat. # CRL-3216 | |
| Cell line (*H. sapiens*) | HUVEC | Lonza | Cat. # C2517A | Primary cell |
| Cell line (*H. sapiens*) | TIME (endothelial cell line) | ATCC | Cat. # CRL-4025 | |
| Cell line (*C. aethiops*) | Vero | CCL-81 | Cat. # CCL-81 | |
| Genetic reagent (*H. sapiens*) | TIME *PCDH1* KO (endothelial cell line) | This study | | Laboratory of K. Chandran/Lentiviral transduction/lentiCRISPR v2-sgRNA *PCDH1* |
| Genetic reagent (*H. sapiens*) | TIME *ITGB3* KO (endothelial cell line) | This study | | Laboratory of K. Chandran/Lentiviral transduction/lentiCRISPR v2-sgRNA *ITGB3* |
| Genetic reagent (*H. sapiens*) | TIME *ITGB1* KO (endothelial cell line) | This study | | Laboratory of K. Chandran/Lentiviral transduction/lentiCRISPR v2-sgRNA *ITGB3* |
| Genetic reagent (*H. sapiens*) | TIME *DAF* KO (endothelial cell line) | This study | | Laboratory of K. Chandran/Lentiviral transduction/lentiCRISPR v2-sgRNA *DAF* |
| Genetic reagent (*H. sapiens*) | TIME *PCDH1/ITGB3* KO (endothelial cell line) | This study | | Laboratory of K. Chandran/TIME *PCDH1* KO + Lentiviral transduction/lentiCRISPR v2-sgRNA *ITGB3* |
| Genetic reagent (*H. sapiens*) | TIME *PCDH1/DAF* KO (endothelial cell line) | This study | | Laboratory of K. Chandran / TIME *PCDH1* KO + Lentiviral transduction/ lentiCRISPR v2-sgRNA *DAF* |
| Genetic reagent (*H. sapiens*) | TIME *ITGB3/DAF* KO (endothelial cell line) | This study | | Laboratory of K. Chandran / TIME *ITGB3* KO + Lentiviral transduction/ lentiCRISPR v2-sgRNA *DAF* |
| Genetic reagent (*H. sapiens*) | TIME *PCDH1* KO + *PCDH1* (endothelial cell line) | This study | Canonical isoform Q08174-1 (Uniprot) | Laboratory of K. Chandran / Retroviral transduction/ pBabe-*PCDH1* |
| Genetic reagent (*H. sapiens*) | TIME *ITGB3* KO + *ITGB3* (endothelial cell line) | This study | Canonical isoform P05106-1 (Uniprot) | Laboratory of K. Chandran / Retroviral transduction/ pBabe-*ITGB3* |
| Genetic reagent (*H. sapiens*) | TIME *DAF* KO + *DAF* (endothelial cell line) | This study | Canonical isoform P08174-1 (Uniprot) | Laboratory of K. Chandran / Retroviral transduction / pBabe-*DAF* |
| Antibody | AF480-α-Human-vWF (Rabbit monoclonal) | Abcam | Cat. # ab195028 | Flow 1:250 |
| Antibody | PeCy7-α-Human-PECAM (Mouse monoclonal) | BD | Cat. # 563651 | Flow 1:1000 |

*Continued on next page*

*Appendix 1—key resources table continued*

| Reagent type (species) or resource | Designation | Source or reference | Identifiers | Additional information |
|---|---|---|---|---|
| Antibody | AF 647-α-Human- β3-Integrin (Mouse monoclonal) | Biolegends | Cat. # 336407 | Flow 1:200 |
| Antibody | PE-α-Human-DAF (Mouse monoclonal) | BD | Cat. # 555694 | Flow 1:1600 |
| Antibody | APC-α-Human-β1-Integrin (Mouse monoclonal) | BD | Cat. # 59883 | Flow 1:20 |
| Antibody | α–Human-PCDH1 mAb 3305 (human monoclonal) | *Jangra et al., 2018* | | Flow 1:200 |
| Antibody | Convalescent serum (human polyclonal) | *Stoltz et al., 2007* | | IF: 1:40 |
| Antibody | AF488-α-–Human IgG (Goat polyclonal) | Invitrogen | Cat. # A-11013 | Flow 1:200 |
| Antibody | AF594-α–Human IgG (Goat polyclonal) | Invitrogen | Cat. # A-11014 | IF 1:500 |
| Antibody | AF594-α-Mouse IgG (Goat polyclonal) | Invitrogen | Cat. # A32742 | IF 1:500 |
| Antibody | AF488-α-Rabbit IgG (Goat polyclonal) | Invitrogen | Cat. # A-11008 | IF 1:500 |
| Antibody | α–Human PCDH1 (Mouse monoclonal) | Santa Cruz | Cat. # sc-81816 | WB 1:200 |
| Antibody | α–Human β3 Integrin (Rabbit polyclonal) | Cell Signaling | Cat. # 4702 | WB 1:300 |
| Antibody | α–Human-DAF (Mouse monoclonal) | Santa Cruz | Cat. # NaM16-4D3 | WB 1:200 |
| Antibody | α–Human β Actin (Mouse monoclonal) | Santa Cruz | Cat. # sc-47778 | WB 1:300 |
| Antibody | IRDye 680LT α-Mouse (Goat polyclonal) | LI-COR | Cat. # 926–68020 | WB 1:10,000 |
| Antibody | IRDye 680LT Goat α-Rabbit IgG 680 (Goat polyclonal) | LI-COR | Cat. # 926–68021 | WB 1:10,000 |
| Antibody | α-hantavirus nucleocapsid B5D9 (Mouse monoclonal) | Progen | Cat. # B5D9-C | IF 1:50 |
| Antibody | α-HTNV nucleoprotein NR-12152 (Rabbit polyclonal) | BEI resources | Cat. # NR-12152 | IF 1:500 |
| Recombinant DNA reagent | pBabe-puro (plasmid) | *Morgenstern and Land, 1990* | Addgene plasmid # 1764 | |
| Recombinant DNA reagent | pBabe-*PCDH1* (plasmid) | *Jangra et al., 2018* | | |
| Recombinant DNA reagent | pBabe-*ITGB3* (plasmid) | This study | | Laboratory of K. Chandran |
| Recombinant DNA reagent | pBabe-*DAF* (plasmid) | This study | | Laboratory of K. Chandran |

*Appendix 1—key resources table continued*

| Reagent type (species) or resource | Designation | Source or reference | Identifiers | Additional information |
|---|---|---|---|---|
| Recombinant DNA reagent | lentiCRISPR v2 (plasmid) | *Sanjana et al., 2014; Shalem et al., 2014* | | |
| Recombinant DNA reagent | lentiCRISPR v2-sgRNA PCDH1 (plasmid) | *Jangra et al., 2018* | | Laboratory of K. Chandran/5'-G TTTGAGCGGCCCTCCTATGAGG-3' / PAM sequence is underlined but not included in oligos |
| Recombinant DNA reagent | lentiCRISPR v2-sgRNA DAF (plasmid) | This study | | Laboratory of K. Chandran / 5'-<u>CCC</u>CCAGATGTACCTAATGCCCA-3' / PAM sequence is underlined but not included in oligos |
| Recombinant DNA reagent | lentiCRISPR v2-sgRNA ITGB3 (plasmid) | This study | | Laboratory of K. Chandran / 5'-<u>CCA</u>CGCGAGGTGTGAGCTCC TGC-3' /PAM sequence is underlined but not included in oligos |
| Recombinant DNA reagent | lentiCRISPR v2-sgRNA ITGB1 (plasmid) | This study | | Laboratory of K. Chandran/5'-AATG TAACCAACCGTAGCAA<u>AGG</u>-3' /PAM sequence is underlined but not included in oligos |
| Sequence-based reagent | SNV degenerate primers | This study | | Laboratory of S. Bradfute/SNV F: CAgC TgTgTCTgCATTggAgAC SNV R: TARAgYCCgATggATTTCCAA TCA |
| Sequence-based reagent | SNV probe | This study | | Laboratory of S. Bradfute/ TMGB1: F-TCAAAACCTgTTgATCCA NFQ MGB |
| Commercial assay or kit | pGEM-T Vector | Promega | Cat. # A3600 | |
| Commercial assay or kit | VIC/TAMRA-dye probe | Applied Biosystems | Cat. # 4310881E | |
| Commercial assay or kit | TaqMan Fast Advanced Master Mix | Applied Biosystems | Cat. # 4444556 | |
| Commercial assay or kit | Foxp3/Transcription Factor Staining | Tonbo | Cat. # TNB-0607-KIT | |
| Commercial assay or kit | Zombie NIR Fixable Viability Kit | BioLegend | Cat. # 423105 | Flow 1:4000 |
| Software, algorithm | Cytation 5 Cell Imaging Multi-Mode Reader | Biotek | | |

