## [Decision Letter]

Thank you for submitting your article "Genetic depletion studies inform receptor usage by virulent hantaviruses in human endothelial cells" for consideration by *eLife*. Your article has been reviewed by 2 peer reviewers, and the evaluation has been overseen by a Reviewing/Senior Editor. The following individual involved in review of your submission has agreed to reveal their identity: Robert Orchard (Reviewer #2).

The reviewers have discussed their reviews with one another, and I have drafted this to help you prepare a revised submission.

Essential revisions:

1) Both reviewers asked you to comment on whether these molecules may have a role in infection of polarized cells, which was the original observation in the literature. While you make a passing mention of this, it is an important point and would be consistent with other viruses (e.g., CVB).

2) Please update the methods to include information on how complementation was performed.

Additional experiments are suggested by the reviewers but will not be required in the revised submission.

*Reviewer #1 (Recommendations for the authors):*

This is an important study that provides insights into the differential effects of host factors on hantavirus infection of endothelial cells. The experiments are rigorously performed and the data directly support the author's conclusions. Appropriate controls are included and the authors are thoughtful in their conclusions.

My only suggestion to the authors would be to consider performing binding assays in the panel of cell lines to determine whether there are alterations at this stage of the viral life cycle by loss of candidate receptors/binding factors. As the authors correctly point out, some viruses utilize attachment factors (including DAF) in a cell-type specific manner and evidence of differential binding might resolve some of the seemingly discrepant data with previous studies.

*Reviewer #2 (Recommendations for the authors):*

Three suggestions for the authors:

1. The authors provide compelling genetic evidence that Integrins and DAF are not required for hantavirus infection of human endothelial cells. Testing if blocking antibodies to B3/B1 integrins and/or DAF alter infection dynamics, may also help clarify any potential role of these molecules to viral infection. This orthogonal assay would bolster the very high quality genetic evidence presented.

2. The authors cite, but do not explain, how Integrins and DAF came be presumed to be involved in hantavirus entry. Given that the assays utilized previously are likely to be quite different than the genetic approach taken by the authors, a brief mention is warranted. This may help put into context any role DAF/Integrin may have on hantavirus infections.

3. I was unable to find information on the complementation of the CRISPR knockout cell lines. Were these complemented using a lentivirus? Was the cDNA altered to prevent sgRNA recognition?

---

## [Author Response]

Essential revisions:1) Both reviewers asked you to comment on whether these molecules may have a role in infection of polarized cells, which was the original observation in the literature. While you make a passing mention of this, it is an important point and would be consistent with other viruses (e.g., CVB).

We have added language to the “Results and Discussion” indicating as an additional caveat that some of the previously proposed receptors could play roles in polarized cell layers, as indeed has been proposed for DAF (Krautkramer and Zeier, 2008).

2) Please update the methods to include information on how complementation was performed.

We thank the reviewer for spotting this omission. The revised manuscript now includes this information in “Materials and methods.”

Additional experiments are suggested by the reviewers but will not be required in the revised submission.

We have also added additional text to “Introduction” to provide context on the experimental approaches used to identify previously proposed receptors, as requested by Reviewer #2.

Reviewer #1 (Recommendations for the authors):This is an important study that provides insights into the differential effects of host factors on hantavirus infection of endothelial cells. The experiments are rigorously performed and the data directly support the author's conclusions. Appropriate controls are included and the authors are thoughtful in their conclusions.My only suggestion to the authors would be to consider performing binding assays in the panel of cell lines to determine whether there are alterations at this stage of the viral life cycle by loss of candidate receptors/binding factors. As the authors correctly point out, some viruses utilize attachment factors (including DAF) in a cell-type specific manner and evidence of differential binding might resolve some of the seemingly discrepant data with previous studies.

We agree that these binding experiments could be informative but chose to focus on the loss-of-function genetic approach for this short report. We are planning to perform such studies to address potential roles for other receptors in specialized settings (such as polarized cells) but note that a failure to demonstrate a direct protein-protein interaction would not preclude the involvement of a candidate molecule as an entry factor.

Reviewer #2 (Recommendations for the authors):Three suggestions for the authors:1. The authors provide compelling genetic evidence that Integrins and DAF are not required for hantavirus infection of human endothelial cells. Testing if blocking antibodies to B3/B1 integrins and/or DAF alter infection dynamics, may also help clarify any potential role of these molecules to viral infection. This orthogonal assay would bolster the very high quality genetic evidence presented.

We chose not to perform such experiments herein so as to focus on a genetic approach that has not been previously explored for these receptor candidates. We agree that blocking experiments would be complementary and help cover more complex scenarios, but these are beyond the scope of this brief report.

2. The authors cite, but do not explain, how Integrins and DAF came be presumed to be involved in hantavirus entry. Given that the assays utilized previously are likely to be quite different than the genetic approach taken by the authors, a brief mention is warranted. This may help put into context any role DAF/Integrin may have on hantavirus infections.

We have also added additional text to “Introduction” to provide context on the experimental approaches used to identify previously proposed receptors.

3. I was unable to find information on the complementation of the CRISPR knockout cell lines. Were these complemented using a lentivirus? Was the cDNA altered to prevent sgRNA recognition?

We thank the reviewer for spotting this omission. The revised manuscript now includes this information in “Materials and methods.” We used codon-optimized versions of the cDNAs for retrovirus-based complementation, which resulted in changes in the sgRNA target sequences (see Author response table 1). Moreover, we were able to demonstrate biochemical complementation of KO under these conditions (see Figure 1C, “+cDNA” lanes).

**Author response table 1. resptable1:** 

	gRNA target site location	Modified sequence in pBabe- *PCDH1/* *ITGB3/ DAF*
*PCDH1*	GTTTGAGCGGCCCTCCTATGAGG	aTTcGAGaGaCCtagtTATGAGG
*ITGB3*	CCACGCGAGGTGTGAGCTCCTGC	CtACtaGAGGcGTatcaagCTGt
*DAF*	CCCCCAGATGTACCTAATGCCCA	CCgCCtGAcGTcCCaAAcGCgCA